# With Greater Text Comes Greater Necessity: Inference-Time Training Helps Long Text Generation

**Yan Wang**[*]  **DM**[*]  **Deng Cai**
Independent Researchers
yanwang.branden@gmail.com, thisisjcykcd@gmail.com

## Abstract

Long text generation, such as novel writing and discourse-level translation with extremely long contexts, presents significant challenges to current language models. Existing methods mainly focus on extending the model's context window through strategies like length extrapolation. However, these approaches demand substantial hardware resources during the training and/or inference phases.

Our proposed method, Temp-Lora, introduces an alternative concept. Instead of relying on the KV cache to store all context information, we embeds this information directly into a temporary Lora module. In the process of long text generation, this module is progressively trained with text generated previously. This approach not only efficiently preserves contextual knowledge but also prevents any permanent alteration to the model's parameters given that the module is discarded post-generation.

Extensive experiments on the PG19 language modeling benchmark and the GuoFeng discourse-level translation benchmark validate the effectiveness of Temp-Lora. Our results show that: 1) Temp-Lora substantially enhances generation quality for long text, as indicated by a **13.2%** decrease in perplexity (PPL) on a subset of PG19, and a **29.3%** decrease in PPL along with a **113.2%** increase in BLEU score on a subset of GuoFeng, 2) Temp-Lora is compatible with and enhances most existing long text generation methods, and 3) Temp-Lora can greatly reduce computational costs by shortening the context window. For example, we can ensure a moderate improvement in generation quality (a decrease of 3.4% in PPL) while enabling a 51.5% memory usage reduction and a 60.0% decrease in latency for inference.

## 1 Introduction

Long text generation has become increasingly important in a variety of real-world applications, ranging from creative writing assistance (Shi et al., 2022), chat-style AI assistant (OpenAI, 2023) to generative agents (Park et al., 2023). However, the generation of coherent and contextually relevant long text poses significant challenges to language models (LMs), particularly in terms of understanding and maintaining contexts that are longer than the model's pre-defined context window size.

Existing methods, including those based on length extrapolation (Press et al., 2022; Su et al., 2023) and context window extension (Chen et al., 2023b; Han et al., 2023; Dao et al., 2022; Peng et al., 2023; Chen et al., 2023a), aims to store extensive text information within the KV cache, thereby improving the model's long text comprehension. However, they demand significant hardware resources during training and/or inference. Consequently, in many applications where LMs are frequently queried for long text processing, users often resort to other strategies such as retrieval or summarization to reduce the cost (Park et al., 2023).

In this paper, we propose an alternative method, Temp-Lora, which stores context information in the model parameters instead of the KV cache. The core idea of our approach is

---

[*] Equal Contribution

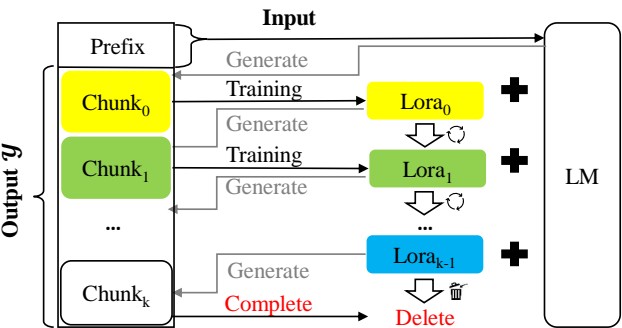

Figure 1: The framework of long text generation with Temp-Lora

extremely simple: we store the context information in a temporary Lora module (Hu et al., 2021) that only exists during long text generation. We update this module in a streaming fashion during the generation process, using the previously-generated content as training data, thereby achieving the goal of storing knowledge in the model parameters. Once inference is complete, this module is discarded to avoid permanently impacting the model's parameters. This approach allows us to efficiently store nearly infinite context information without extending the context window.

We evaluate Temp-Lora on two benchmark datasets, PG19 (Rae et al., 2019) and GuoFeng (Wang et al., 2023b). We evaluate Temp-Lora across models with different context window sizes and find that Temp-Lora substantially reduces their PPL on long text with a large margin (-13.2% on PG19 and -29.3% on GuoFeng). This trend becomes increasingly pronounced as the context length increases, which can be simply concluded as "**with greater text comes greater necessity**". Further analysis also reveals that Temp-Lora, by shortening the maximum input length, can greatly reduce hardware consumption.

## 2 Temp-Lora

The Temp-Lora framework, depicted in Figure 1, presents a straightforward yet innovative approach for long text generation. At the heart of this method lies the progressive training of a temporary Lora module (Hu et al., 2021), which is named Temp-Lora, on previously-generated text in an auto-regressive manner. The continuous adaptation and refinement of the Temp-Lora module ensure an evolving understanding of both recent and distant contexts. It is important to highlight the framework's adaptability in handling cases where the initial input lengths are already substantial, such as novels or academic papers. In such scenarios, we may proactively pre-train the Temp-Lora module with the contexts, thereby laying a robust foundation for enhanced context comprehension in the generation process.

The algorithm is detailed in Algorithm 1. During the generation process, tokens are generated chunk-by-chunk. At each chunk generation, we use **the most recent $L_{\mathcal{X}}$ tokens** as the input $\mathcal{X}$ for generating the subsequent tokens (all preceding tokens beyond the distance $L_{\mathcal{X}}$ are discarded). Once the number of generated tokens hits a predefined **chunk size $\Delta$**, we initiate the training of the Temp-Lora module using the latest chunk and start the next chunk generation afterwards. It is important to note that $L_{\mathcal{X}} + \Delta \leq \mathcal{W}$, where $\mathcal{W}$ is the model's context window size.

For the training of the Temp-Lora module, it is crucial to recognize that learning to generate the new chunk without any condition may not constitute a meaningful training objective and potentially lead to significant over-fitting. To address this concern, we incorporate the preceding $L_T$ tokens of each chunk into our training process, using them as the input and the chunk as the output.

**Cache Reuse** For more efficient inference, we propose a strategy called **cache reuse**. In the standard framework, after each update of the Temp-Lora module, we need to re-compute

---

**Algorithm 1:** Long Text Generation with Temp-Lora

---

**Input:** Input $\mathcal{X}$; Input Length $L_{\mathcal{X}}$; Training Input $\mathcal{X}_T$; Training Input Length $L_T$; Chunk size $\Delta$; Learning rate $\alpha$; Epoch number $n$; LM $\Theta$.
**Output:** Long sequence $\mathcal{Y}$

1 **if** $len(\mathcal{X}) > L_{\mathcal{X}}$ **then**
2    |  Pre-train $\Theta$ on $\mathcal{X}$ to obtain LM with an initial Temp-Lora module $\Theta_0$;
3 **end**
4 Total token number: $i \leftarrow len(\mathcal{X})$;
5 Temp-Lora ID: $k \leftarrow 0$;
6 Token number in this chunk: $m \leftarrow 0$;
7 Output sequence $\mathcal{Y} = \mathcal{X}$;
8 $\mathcal{X} = \mathcal{X}[-(\min(L_{\mathcal{X}}, len(\mathcal{X})) :]$;
9 **while** *In Generation* **do**
10   | **if** $m < \Delta$ **then**
11   |   | $\mathcal{Y}[i] = \Theta_k(\mathcal{X} + \mathcal{Y}[-m :])$;
12   |   | $i + +; m + +$;
13   | **end**
14   | **else**
15   |   | $Chunk_k = \mathcal{Y}[-m :]$;
16   |   | $\mathcal{X}_T = \mathcal{Y}[-(L_T + m) : -m]$;
17   |   | $k + +$;
18   |   | Train $\Theta_k$: In $\mathcal{X}_T \rightarrow$ Out $Chunk_k$ (n epochs, learning rate $\alpha$);
19   |   | $\mathcal{X} = \mathcal{Y}[-L_{\mathcal{X}} :]; m = 0$;
20   | **end**
21 **end**
22 Destroy Temp-Lora module: $\Theta_k \leftarrow \Theta$;

---

the KV states of the latest $L_{\mathcal{X}}$ tokens with the updated parameters. Alternatively, we can also reuse the existing cached KV states while employing the updated model for subsequent text generation. We empirically find that cache reuse can accelerate the generation speed without compromising the generation quality.

**Attention Sinks** The recently proposed Attention Sink (Xiao et al., 2024) method allows us to further enhance inference efficiency. By permanently storing four initial tokens as attention sinks in the KV cache, the model, after generating up to the maximum length, no longer needs to recompute the KV states. Therefore, if we simultaneously employ cache reuse and attention sink strategies, during long-text inference, we only need to continuously update the model (Temp-Lora module) without re-computing the KV states.[1]

## 3 Experiments

We evaluate the proposed Temp-Lora framework using the Llama2 (Touvron et al., 2023) families and Yi-Chat-6B considering their wide adoption and popularity. Its effectiveness is evaluated in two different long text generation tasks: 1) Novel Generation; and 2) Discourse-Level Literary Translation.[2]

**Dataset:** The first dataset we adopt is a subset of the long text language modeling benchmark, PG19 (Rae et al., 2019). It is a well-established benchmark that consists of more than 28K books which were published before 1919. Since we are especially concerned with the effectiveness of the Temp-Lora framework in extremely long text scenarios, and considering that some of the PG-19 data might already be included in our models' pre-training corpora,

---

[1]Please note that simply adding attention sink without implementing cache reuse is ineffective because, without cache reuse, each update of Temp-Lora necessitates re-computing KV states
[2]Codes and Data are available at: https://github.com/TemporaryLoRA/Temp-LoRA/tree/main

we select 100 books with the highest PPL from those whose lengths range between 200K and 600K tokens. From these, we randomly sample 40 books as our test set.

We also evaluate the effectiveness of Temp-Lora on a downstream task, Discourse-Level Literary Translation, with a randomly sampled subset of GuoFeng dataset from WMT 2023 (Wang et al., 2023b;a). This subset contains 20 web novels, originally written in Chinese by various novelists and subsequently translated into English by professional translators. Their lengths (en + zh) range from a minimum of 84K to a maximum of 370K tokens.

We split novels in the GuoFeng dataset into segments based on sentence boundaries. Each segment has approximately 512 tokens (en + zh) in terms of length. The model's input includes the current source segment along with its historic source and target text, and it is required to output the translation of current source segment.

**Baselines:** We apply the Temp-Lora framework to three Llama2 variants and Yi-Chat-6B:

- **Llama2-7B-4K**: the standard Llama2-7B model. [3]
- **Llama2-13B-4K**: the standard Llama2-13B model. [4]
- **To-Llama2-13B-32K**: This is a very strong baseline model that extends the context window of Llama2-7B-4K to 32K. The developers from TogetherAI followed Meta's linear interpolation (Chen et al., 2023b) and continue pre-trained it with 1.5B tokens. We believe it is **much stronger than those training-free methods**. [5]
- **Mistral-7B**: the standard Mistral-7B-v0.1 model with a 8K context window size. [6]
- **phi-2**: the standard phi-2 model with a 2K context window size. [7]
- **qwen-7B**: the standard qwen-7B model with a 8K context window size. [8]
- **Yi-Chat-6B**: A chat-style model trained on both Chinese and English corpus. We will use this model in the Discourse-Level Literary Translation task. [9].

Please note that the length of most samples in our test set significantly surpass the models' context window sizes. In such instances, when the model hits the token generation limit, we slide the context window and recompute the KV states from the recent $\mathcal{W} - 1024$ tokens.

**Evaluation Metric:** The primary metric is perplexity (PPL), a standard measure in language modeling to assess the model's prediction capability. We employ a sliding window approach for PPL measurement as suggested by (Press et al., 2022). For the translation task, in addition to PPL, we also employ two common evaluation metrics used in machine translation, BLEU (Papineni et al., 2002) and COMET (Rei et al., 2020), for comprehensive evaluation.

**Setup:** We set the chunk size $\Delta = 1024$ for all models. For the Llama2-7B-32K, we additionally set two wider chunk size $\Delta = 2048$ and $4096$ to investigate the effects of chunk size on generation quality and computational cost.

The input length $L_\mathcal{X}$ for all models is set to be $\mathcal{W} - \Delta$ for maximizing the use of the models' context window size. For the Llama2-7B-32K, we notice that its PPL increases when the number of tokens within its context window approaches 32K. Therefore, we set its context window size $\mathcal{W}$ to 24K, which, based on our preliminary experiments, is the optimal context window size. Note that the number of context tokens may be smaller than $L_\mathcal{X}$. In such cases, we simply take all context as the input. In the generation process, by default, once the token number in the context window reaches the context window size (4K and 24K for different

---

[3] https://huggingface.co/meta-llama/Llama-2-7b
[4] https://huggingface.co/meta-llama/Llama-2-13b
[5] https://huggingface.co/togethercomputer/LLaMA-2-7B-32K
[6] https://huggingface.co/mistralai/Mistral-7B-v0.1/discussions/4
[7] https://huggingface.co/microsoft/phi-2
[8] https://huggingface.co/Qwen/Qwen-7B
[9] https://huggingface.co/01-ai/Yi-6B-Chat

models) we re-compute the KV states taking the $L_X$ recent tokens as input $X$ (i.e., no cache reuse). When we update the Temp-Lora module, the length $L_T$ is set to 1024.

In the experiments on GuoFeng, each segment is treated as a chunk. In other words, after the model translates a complete segment, this segment is then used to update the Temp-Lora module. We decode the translations with greedy search (repetition penalty = 1.12). [10]

All experiments are done with a single NVIDIA A800 GPU. We set the learning rate $\alpha = 5 \times 10^{-5}$ and the number of epochs $n = 2$ for all models. We use a linear learning rate warmup for the first 2 chunks. We set Temp-Lora $\alpha = 64$, $rank = 64$, $dropout = 0.05$. Its training is in bfloat16 format, using Deepspeed ZeRO Stage 2, Flash Attention V2 (Dao, 2023), and gradient checkpointing.

| Model | $\Delta$ | 0-100K | 100-300K | 300-500K | 500K+ | Avg. |
|---|---|---|---|---|---|---|
| **Llama2-7B-4K** | - | 10.14 | 9.76 | 8.88 | 4.57 | 9.81 |
| +TL | 1024 | 9.79 (3.4%) | 9.07 (7.0%) | 8.07(9.1%) | 3.96 (13.2%) | 9.23 (5.9%) |
| **Llama2-13B-4K** | - | 9.21 | 8.88 | 8.12 | 4.19 | 8.93 |
| +TL | 1024 | 8.91 (3.1%) | 8.31 (6.3%) | 7.44 (8.4%) | 3.70 (11.6%) | 8.45 (5.3%) |
| **Mistral-7B** | - | 10.58 | 10.16 | 8.83 | 4.32 | 10.20 |
| +TL | 1024 | 10.34 (2.2%) | 9.57 (5.7%) | 8.19 (7.1%) | 3.74 (13.3%) | 9.73 (4.5%) |
| **Qwen-7B** | - | 18.39 | 17.51 | 14.08 | - | 17.70 |
| +TL | 1024 | 18.02 (2.0%) | 16.44 (6.1%) | 13.05 (7.3%) | - | 16.90 (4.5%) |
| **To-Llama2-7B-32K** | - | 10.21 | 9.74 | 8.70 | 4.01 | 9.81 |
| +TL | 1024 | 9.92 (2.8%) | 9.15 (6.0%) | 7.96 (8.4%) | 3.66 (8.8%) | 9.32 (5.0%) |
| +TL + CR | 1024 | 9.94 (2.7%) | 9.16 (6.0%) | 7.97 (8.3%) | 3.68 (8.3%) | 9.33 (4.9%) |
| +TL | 2048 | 9.95 (2.6%) | 9.17 (5.8%) | 7.99 (8.1%) | 3.71 (7.6%) | 9.34 (4.8%) |
| +TL + CR | 2048 | 9.97 (2.4%) | 9.18 (5.7%) | 8.01 (7.9%) | 3.75 (6.6%) | 9.36 (4.7%) |
| +TL | 4096 | 9.99 (2.1%) | 9.21 (5.4%) | 8.04 (7.6%) | 3.77 (6.1%) | 9.38 (4.4%) |
| +TL + CR | 4096 | 10.03 (1.8%) | 9.23 (5.2%) | 8.06 (7.3%) | 3.81 (5.0%) | 9.41 (4.1%) |
| **To-Llama2-7B-32K + AS** | - | 10.35 | 9.91 | 8.88 | 4.20 | 9.98 |
| +TL + CR | 1024 | 10.02 (3.1%) | 9.26 (6.6%) | 8.08 (9.0%) | 3.78 (9.8%) | 9.42 (5.5%) |
| +TL + CR | 2048 | 10.05 (2.8%) | 9.28 (6.3%) | 8.12 (8.6%) | 3.85 (8.3%) | 9.45 (5.2%) |
| +TL + CR | 4096 | 10.12 (2.2%) | 9.34 (5.8%) | 8.18 (7.9%) | 3.93 (6.4%) | 9.51 (4.6%) |

Table 1: PPL of different moedls. All tokens in PG19 are divided into four segments based on their position in the novel, i.e., the actual context length of the token. For instance, if a token is the 160K-th token in a novel, it would be categorized into the 100-300K segment. In this table, we report the PPL of various models under different settings. The percentages in () indicate the relative reduction of PPL of the model compared to the base model. TL, CR, and AS are short for Temp-Lora, cache reuse, and attention sink, respectively.

## 3.1 Main Results

**PG19:** Table 1 presents the PPL comparisons across various models with and without the Temp-Lora module on PG19. We measure the PPL scores at different positions separately. Specifically, for a sequence x, we compute the PPLs of the following subsequences: $[x[:100K], x[100K:300K], x[300K:500K], x[500K:]]$. Intuitively, In the initial segments such as 0-100K, where there is less context information, the improvement from Temp-Lora should be relatively modest. In contrast, as the segments grow longer and more information falls outside the model's context window, Temp-Lora's effects should become more pronounced.

The experimental results in Table 1 confirm our hypothesis. Firstly, the augmentation of Temp-Lora leads to a significant PPL reduction for all models, where we observe an average decrease of 5.9% on Llama2-7B-4K. An in-depth examination of Temp-Lora's impact across different text segments reveals it is more notable on long text. For example, Temp-Lora augmented Llama-7B-4K achieves a direct PPL reduction of **13.2%** in the 500K+ segment.

---

[10] For the base model Yi-Chat-6B, we have included the most recent three segments as examples in the input to better maintain consistency in translation; for the Temp-Lora augmented model, there is no need to retain examples in the input

| Model | zh2en | | | en2zh | | |
|---|---|---|---|---|---|---|
| | PPL↓ | BLEU↑ | COMET↑ | PPL↓ | BLEU↑ | COMET↑ |
| Yi-6B-Chat | 4.36 | 12.4 | 72.5 | 6.99 | 11.5 | 77.2 |
| +TL | 3.32 (-23.8%) | 19.0 (53.2%) | 78.6 (8.4%) | 4.95 (-29.1%) | 24.7 (113.1%) | 82.8 (7.2%) |

Table 2: PPL, BLEU, and COMET of Yi-6B-Chat with and without Temp-Lora.

Surprisingly, on segments whose context length is greater than 300K, Temp-Lora helps Llama2-7B achieve a lower PPL than the 13B model. In contrast, its effect is relatively less pronounced in the 0-100K segment, where it only reduces the PPL by 3.6%. We may simply conclude the results as: **with greater text comes greater necessity for Temp-Lora**.

Experiments with the strong baseline model, To-Llama2-7B-32K, which was fine-tuned on 1.5 billion tokens of long-context language data, demonstrate that **Temp-Lora operates orthogonally to existing long text generation techniques**. Even though this model has already employed the most advanced Context Window Extension and Length Extrapolation techniques, and continue pre-trained with 1.5B tokens, within the Temp-Lora augmentation, it achieves a reduction in Perplexity (PPL) comparable to that observed in those 4K and 8K context models.

Furthermore, adjusting the chunk size from 1024 to 2048 and 4096 resulted in a slight increase in PPL. It is not surprising, as the Temp-Lora module was trained on the data from the previous chunks. An increase in chunk size means that the information stored in the module is more distant from the current token. Indeed, the choice of chunk size is a critical trade-off between generation quality and computational efficiency, which will receive detailed analysis in Section 3.2.

**Cache reuse and attention sink:** We also discover that cache reuse almost does not cause any loss in performance. This is a very encouraging observation, as this technique can significantly reduce computational cost. Moreover, another set of experiments verifies the compatibility of Temp-Lora with the attention sink. Encouragingly, the experimental results confirmed that these two can be seamlessly integrated. With the augmentation of Temp-Lora, the reduction in PPL of Llama2-7B-32K + attention sink is even greater than its standard version (5.5% vs 4.9%, 5.2% vs 4.7%, and 4.6% vs 4.1%).

**GuoFeng:** Table 2 presents the remarkable impact of the Temp-Lora on the task of discourse-level literary translation. Compared to the base model Yi-Chat-6B, there are significant improvements across all metrics and all language pairs: a decrease in PPL by **-23.8%**, an increase in BLEU score by **+53.2%**, and a rise in COMET score by **+8.4%** in Chinese-to-English translation; a decrease in PPL by **-29.1%**, an increase in BLEU score by **+113.1%**, and a rise in COMET score by **+7.2%** in English-to-Chinese translation. This experiment illustrates that the Temp-Lora module is also effective in downstream tasks.

One might wonder why the effects are more pronounced in translation. This is attributed to the differences between tasks: open-domain novel completion is a task without standard answers, making it challenging for even an active human reader of a specific novel to predict the content of the next chapter. In such cases, even if the Temp-Lora module stores substantial context information, the model can only slightly reduce the PPL of ground-truth text. In contrast, in discourse-level translation, most word translations are unique for consistency. By effectively storing these translation mappings in the Temp-Lora module, substantial improvements can be readily achieved.

## 3.2 Further Analysis

**Contrast to Dynamic-NTK** One might be curious about how the existing Length Interpolation methods perform in these super-long context scenarios. Table 3 presents the outcomes on the PG19 dataset after extending the Llama2-7B-4K model using Dynamic-NTK (NTK, 2023; Peng et al., 2023). Unfortunally, Dynamic-NTK are not suitable for this scenario. One may easily find that once the context window extends to more than four times its training window, PPL will collapse directly.

| Size | 0-100K | 100-300K | 300-500K | 500K+ | Avg. |
|------|--------|----------|----------|-------|------|
| 8K   | 11.2   | 10.9     | 9.7      | 5.3   | 10.9 |
| 16K  | 22.5   | 24.1     | 20.9     | 10.5  | 23.2 |
| 24K  | 65.8   | 87.0     | 78.3     | 24.8  | 77.7 |
| 32K  | 106.3  | 161.0    | 147.1    | 63.3  | 137.1 |
| 40K  | 212.4  | 420.6    | 385.7    | 145.1 | 324.4 |

Table 3: PPL of Llama2-7B-4K with Dynamic-NTK

**Context Window Size:** In Section 3.1, we demonstrate the Temp-Lora framework's ability to enhance long text generation. A key observation is that certain context information is simultaneously stored within both the model's parameters (Temp-Lora) and its KV cache, leading to an overlap. For a further understanding of the framework's efficiency, we gradually shorten the context window size $\mathcal{W}$ during inference to eliminate this overlap. After shortening $\mathcal{W}$, we can reduce the computational cost for both models. However, this reduction results in less context information being stored in the KV cache, compelling the model to rely primarily on the Temp-Lora module for accessing contextual information. We measure Llama-7B-32K's PPL, memory usage, and latency across these varied context window sizes $\mathcal{W}$ in Table 4. This experiment aims to explore the balance between generation quality and computational efficiency in different scenarios.

We observe that models augmented with Temp-Lora consistently surpass the base model's performance, regardless of the context window size. Notably, even when we reduce the window size to **1/12** of its maximum (2048), a reduction in PPL from 9.81 to 9.65 is also observed. Please note that as the context window diminishes, there is still a gradual increase in the model's PPL. As a whole, this trend highlights that Temp-Lora and the KV cache are orthogonal, jointly enhancing the overall performance of the model when used together.

| Metric | Δ | Base | 24K | 16K | 8K | 4K | 2K |
|--------|---|------|-----|-----|-----|-----|-----|
| PPL | 1024 | 9.81 | 9.32 (-5.0%) | 9.33 (-4.9%) | 9.39 (-4.3%) | 9.48 (-3.4%) | 9.65 (-1.6%) |
| PPL | 2048 | 9.81 | 9.34 (-4.8%) | 9.36 (-4.6%) | 9.43 (-4.0%) | 9.55 (-2.7%) | - |
| PPL | 4096 | 9.81 | 9.38 (-4.3%) | 9.41 (-4.1%) | 9.50 (-3.1%) | - | - |
| Mem. (GB) | - | 38.5 | 38.8 (+0.3%) | 30.5 (-20.6%) | 22.6 (-41.2%) | 18.6 (-51.5%) | 16.7 (-56.5%) |
| Lat.(s) | - | 60.1 | 60.2 (+0.1%) | 44.4 (-26.0%) | 28.8 (-52.0%) | 24.0 (-60.0%) | 23.9 (-60.1%) |

Table 4: PPL, Memory Usage, and latency (1K tokens) across various context window sizes $\mathcal{W}$ for To-Llama2-7B-32K with Temp-Lora and cache reuse. The base model's context window size is 24K. Tokens are decoded with greedy search. Note that $\mathcal{W}$ should be larger or equal to the sum of the chunk size and training input length, $\Delta + L_T$. If not, the context window will be in-sufficient to include all tokens in Temp-Lora updating. The percentages in () indicate the relative PPL change of the model compared to the base model's best.

| Δ | Metric | 4096 | 2048 | 1024 | 512 | 256 |
|---|--------|------|------|------|-----|-----|
| 1024 | Lat (s) | - | - | 1.67 | 2.38 | 3.83 |
|      | Mem. (GB) | - | - | 18.0 | 17.4 | 17.3 |
| 2048 | Lat (s) | - | 2.44 | 4.28 | 5.35 | 7.40 |
|      | Mem. (GB) | - | 19.0 | 18.0 | 17.5 | 17.3 |
| 4096 | Lat (s) | 3.59 | 4.97 | 5.88 | 8.46 | 14.02 |
|      | Mem. (GB) | 21.0 | 19.0 | 18.0 | 17.5 | 17.3 |

Table 5: Cost of Temp-Lora training for To-Llama2-7B-32K. The numbers in the first row represent the batch size during training. For example, if we set the chunk size $\Delta = 2048$, the batch size as 1024, and training epochs $n = 2$, then updating the Temp-Lora module requires $2 * (2048/1024) = 4$ training steps.

**Efficiency:** Next, let's explore the efficiency of the Temp-Lora-augmented model in practice. When deploying the model in real-world applications, two distinct deployment strate-

gies can be considered based on the availability of computational resources: 1) **Cascaded:** As outlined in Algorithm 1, this strategy temporarily pauses inference after generating a chunk of data. This pause allows for the Temp-Lora module to be updated with the newly generated chunk before resuming the generation process. 2) **Parallelized:** When abundant computational resources are available, the Temp-Lora module can be updated in parallel with the token generation process during inference. In this approach, after generating a chunk, we do not pause inference. Instead, we update the Temp-Lora module using this chunk in another process. Once the Temp-Lora module is updated, we immediately replace the old one with the updated version.

With the **parallelized deployment** strategy, the model's memory usage and inference latency during inference, as shown in Table 4 are nearly identical to those of the base model with the same context window size. We were surprised to find that in the most "latency-sensitive" scenario, we may set $\Delta = 1K$ and $\mathcal{W} = 4K$, the model **not only reduces PPL by 3.4% but also saves 51.5% of Memory usage and 60.0% of latency**. Conversely, if we disregard computational costs entirely, then in the most "luxurious" configuration, $\Delta = 1K$ and $\mathcal{W} = 24K$, we can achieve a 5.0% reduction in PPL with minimal additional memory consumption (0.3 Gb) and latency (0.1 ms).

On the other hand, in scenarios of limited resources, the **cascaded deployment strategy** becomes preferable. In such cases, the total time required to generate a chunk equates to the sum of the model's inference latency and the time spent updating the Temp-Lora. As detailed in Table 5, we demonstrate the relationship between training latency and memory usage across various training batch sizes. Notably, training Temp-Lora is significantly faster than the inference process itself. For instance, in the "latency-sensitive" configuration, with $\Delta = 1K$ and $\mathcal{W} = 4K$, the average inference latency for every 1K tokens is 24 seconds, whereas Temp-Lora training takes merely 1.67 seconds. Furthermore, regarding memory usage, it is reassuring that the memory required for Temp-Lora training rarely surpasses that during inference. Thus, repurposing the memory allocated for inference to train the Temp-Lora module is viable, eliminating the risk of an 'Out of Memory' issue.

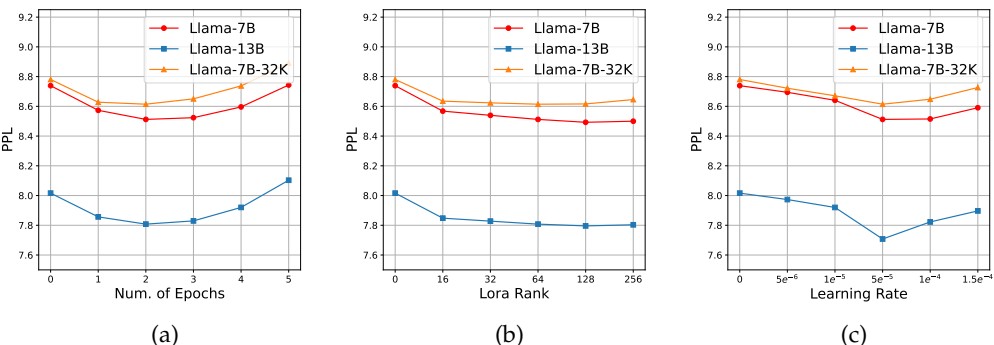

Figure 2: The relationship between model PPL and hyper-parameters, depicted from left to right, are (a) Epochs, (b) Lora Rank, and (c) Learning Rate. Note that the PPL at a X-axis value of 0 in each graph corresponds to the performance of the base model.

## 3.3 Discussion

One might find it surprising that the training cost for Temp-Lora is significantly lower than for inference, despite the widespread belief that model training generally requires more time than inference. This efficiency stems from the model's ability to leverage parallel processing techniques during the training phase, allowing it to handle the entire chunk simultaneously. In contrast, the generation phase is constrained by the autoregressive nature of the language model, necessitating a token-by-token generation approach. This fundamental difference significantly accelerates the training of a chunk compared to its generation.

After summarizing the experimental results, we got some suggestions for applying Temp-Lora in real-world scenarios: 1) For applications demanding the highest level of long text generation, integrating Temp-Lora into existing models — without altering any parameters — can significantly enhance performance at a relatively modest cost. 2) For applications where minimal latency or memory usage is paramount, computational costs can be significantly lowered by reducing input lengths and storing context information within Temp-Lora. Under this setup, we can process text of almost infinite length (500K+ in our experiments) using a fixed short window size. 3) Note that in scenarios without extensive text, for example, less than the model's window size in training, Temp-Lora is useless.

### 3.4 Parameter Sensitivity Analysis

One may wonder how sensitive Temp-Lora is to the training hyper-parameters. In Figure 2, we investigate the sensitivity of our model to three key hyper-parameters, Epochs, Lora Rank, and Learning Rate, on a subset of PG19 that contains 3 books.

**Epochs:** training each chunk for just one epoch consistently leads to a significant PPL reduction, where we observe an average decrease of about **0.2**. Expanding the number of epochs to two further reduces PPL, but excessively high epoch counts risk catastrophic forgetting, and negating the benefits by losing previously learned information.

**Lora Rank:** lora rank determines the number of parameters in Temp-Lora module. From Figure 2 (b) we observe that the model is robust to this hyper-parameter. When the lora rank increases from 16 to 256, the model's PPL remains essentially stable.

**Learning Rate:** from Figure 2 (c) we observe that the learning rate is a relatively important parameter that has a significant impact on the model's performance. Intuitively, a too-small learning rate may lead to less knowledge being stored in the Temp-Lora module, whereas a too-large learning rate might cause the model to overfit to the current chunk.

## 4 Related Work

In recent years, numerous efforts have been made to enable language models to understand and generate longer texts (Pawar et al., 2024; Zhao et al., 2023). The primary focus of these studies has been to store more information within the context window. They can be broadly categorized into three types: 1) Length Extrapolation, 2) Context Window Extension, 3) External Memory. It is worth noting that there are some overlaps between categories 1) and 2). Following Chen et al. (2023b), we categorize training-free methods under Length extrapolation, while those fine-tune-based methods are categorized under context window extension. Our Temp-Lora framework pioneers a new direction in long-text generation area, which is orthogonal to these three types and can be seamlessly integrated.

**Length Extrapolation** aims to find ways to process long contexts with short context windows. This "Train Short, Test Long" paradigm was first introduced in Press et al. (2022), which proposed the ALiBi position embedding method that leverages linear-decaying attention biases to achieve the extrapolation of position encoding. On the top of another position embedding method, RoPE (Su et al., 2023), Methods like Position Interpolation (Chen et al., 2023b), NTK-Aware scaling (NTK, 2023), and ReRoPE (Su, 2023) make some further progresses to enable the LLM to handle unseen positions at the inference time. A recent method, SelfExtend (Jin et al., 2024), attempt to extend the context window of LLMs by constructing bi-level attention information. Based on some empirical experience, although these methods can theoretically be scaled to an infinite context window length, in practical applications, extending the context window length to more than four times the size of the context window will lead to a serious degradation problem.

**Context Window Extension** centers on expanding the LLMs' context window, enabling the processing of more tokens in one forward pass. The principle behind "Extending" might sound straightforward: train LLMs on longer texts. However, the most significant challenge lies in the training efficiency problem. Consequently, researchers are exploring ways to reduce the training costs. Their solutions range from system-focused optimizations like

FlashAttention (Dao et al., 2022; Dao, 2023) that accelerates attention computation and reduces memory usage, to approximate attention methods (Zaheer et al., 2021; Beltagy et al., 2020; Wang et al., 2020; Kitaev et al., 2020) that trade model quality for efficiency. Additionally, Most Length Extrapolation methods (Chen et al., 2023b; Peng et al., 2023; Zhu et al., 2024) may also benefit from continual fine-tuning, where the extended context can be better utilized. Beyond algorithm-level optimization, ProLong (Chen et al., 2024a) try to tackle this long-context modeling issue from a data-centric perspective, which only filters a small amount of long dependency data for extension training. Recently, researchers also leveraged LoRA for cost-effective training (Chen et al., 2024b). However, all these techniques only extend LLMs' context window to a limited extent, which falls short of our paper's primary concern of handling infinite inputs.

**External Memory** tackles the long-context understanding problem from a different perspective: It stores all necessary knowledge into a pre-computed index and only retrieves useful data as the working context (Li et al., 2022). The retrieved data can be either the supervised data (Cai et al., 2021) or chunked text (Lan et al., 2023; Xu et al., 2024). Although the performance of Retrieval-Augmented Generation (RAG) continues to improve (Mohtashami & Jaggi, 2023; Tworkowski et al., 2023), its biggest challenges lie in 1) the accuracy of retrieval is still far from satisfactory, and 2) incorrect retrieval can lead to error accumulation, resulting in a decline in the quality of the generated content.

**Concurrent Work** Our research coincides with the work of Sun et al. (2024), which proposed to add a Test-Time Training (TTT) layer to the model to store the long context. The main difference between our work and theirs is that we focus on the Transformer architecture, while they primarily concentrate on the RNN architecture.

## 5 Conclusion

We proposed Temp-Lora, an alternative way for efficient long text generation. Its essence lies in training during the inference process using the generated output. It enables the storage of nearly infinite context information directly within the model's parameters, marking a distinct difference from existing attention weights-based techniques.

Our experimental results across various applications, including language modeling and discourse-level literary translation, demonstrated the profound impact of Temp-Lora. We showed that Temp-Lora not only greatly enhances the quality of long text generation but also significantly reduces computational costs. In particular, Temp-Lora led to remarkable improvements in perplexity (a reduction of 13.2% on a subset of PG19 and 29.3% on a subset of GuoFeng), as well as increases in BLEU (by 113.2%). The effectiveness of Temp-Lora becomes increasingly apparent as text length grows. When considering the implementation of Temp-Lora in your applications, bear in mind this guiding principle: **With Greater Text Comes Greater Necessity for Temp-Lora** – a rule that becomes increasingly relevant in the context of extremely long text.

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
