# OpenReview forum: "With Greater Text Comes Greater Necessity: Inference-Time Training Helps Long Text Generation"
_colmweb.org/COLM/2024/Conference — COLM_

### Official Review · Reviewer_535h · 2024-04-15

**Rating:** 5
**Confidence:** 5
**Ethics Flag:** 1

**Summary:**

This paper introduce a simple and effective framework Temp-Lora to extend the context window for LLMs for long text generation.
The author argues that traditional methods (length extrapolation & context window extension) demand significant hardware resources
during training / inference,  so Temp-lora is proposed to store context information in model parameters instead of KC cache.
The parameters of Temp-Lora is training on the fly using previously-generated tokens, and discarded when the inference is completed.
Experiment on PG19 data-set shows a significant improvement in PPL/BLEU, and a large amount of memory reduction and latency decrease.

**Questions To Authors:**

see weakness part.
Also in Figure 1, there is a missing arrow to point to Chunk1 to "Generate"

**Reasons To Accept:**

The motivation of this paper is solid, focusing resolving the context limitation for long text generation.
The proposed method is simple and effective, the paper is easy to understand.
And the experimental results seems promising, achieving PPL reduction, memory reduction & latency reduction at the same time.

**Reasons To Reject:**

1. I think the related work comparison is weak. In the intro, the author strongly claims that "Temp-Lora is compatible with and enhances most existing long text generation methods."  But I don't see enough evidences in the experiment part to support this claim.  So maybe the author can elaborate on this point.
2. It's exciting to see good experimental results  (achieving PPL reduction, memory reduction & latency reduction at the same time),
but on the other side of the coin, I think it's also partially because the baseline is relatively weak, maybe adding more comparison system with the length extrapolation and context window extension methods will make the results more convincing.
3. One of the main advantage of this method is preserving the hardware resources, I assume it also includes computation cost,
and the on-the-fly training of Temp-lora requires additional GPU during inference.  So maybe the author could add more detail on this direction, and list out computation overhead for a better presentation of this paper.

---

> ### Author Rebuttal · Authors · 2024-05-29
>
> **Concern 1 & 2**:
> Thanks for your valuable suggestions! Firstly we should clarify that we already compare with a
> **very strong baseline "Together LLaMA2-7B-32k"**
> in our paper. Please refer to our response to **Conern 1 from Reviewer 3Zws**, for more details of this strong baseline, and the comparision with other existing baselines.
>
> Furthermore, please refer the response to **Concern 1 from Reviewer Sdmf** for details about the evaluation of the proposed method on other LLMs.
>
> Finally, we also verify the compatibility of Temp-Lora with the StreamingLLM method (Efficient Streaming Language Models with Attention Sinks"). The model "7B-32K+AS" in Table 1 represents this method. This is our analysis in Section 3.1:
>
> Moreover, another set of experiments verifies the compatibility of Temp-Lora with the recent proposed StreamingLLM method (attention sink). Encouragingly, the experimental results confirmed that these two can be seamlessly integrated. With the augmentation of Temp-Lora, the reduction in perplexity (PPL) of Llama2-7B-32K + attention sink is even greater than its standard version (5.5% vs 4.9%, 5.2% vs 4.7%, and 4.6% vs 4.1%).
>
> **Concern 3**: maybe the author could add more detail on the computation cost, and list out computation overhead for a better presentation of this paper
>
> In fact, we have already conducted an analysis in this regard in Section 3.2, and we will elaborate it more clearly in the next version.
>
> In Table 4, we detailed the computational cost of on-the-fly training of the Temp-Lora module under different chunk sizes and batch size settings. By comparing Table 3 and Table 4, we can conclude that: Under similar memory usage conditions, the time consumed to update the Temp-Lora module with one chunk is only 1/15 of the time used to infer this chunk (1.67s vs 24s). The analysis in our paper is as follows:
>
> Notably, training Temp-Lora is significantly faster than the inference process itself. For instance, in the 'latency-sensitive' configuration, with ∆ = 1K and W = 4K, the average inference latency for every 1K tokens is 24 seconds, whereas Temp-Lora training takes merely 1.67 seconds. Furthermore, regarding memory usage, it is reassuring that the memory required for Temp-Lora training rarely surpasses that during inference. Thus, repurposing the memory allocated for inference to train the Temp-Lora module is viable, eliminating the risk of an 'Out of Memory' issue.

---

### Official Review · Reviewer_1qNy · 2024-05-10

**Rating:** 6
**Confidence:** 3
**Ethics Flag:** 1

**Summary:**

This paper propose a method called Temp-Lora, which embeds the context information into a Lora module for each chunk to enable long text generation. The proposed method is implemented based on Llama2 and evaluated on two benchmark datasets. The performance on PPL, memory usage and latency get improved compared with Llama2.

**Questions To Authors:**

Can the author show some cases generated by the proposed method? Only providing the PPL can't intuitively show the generation performance of the proposed method.

**Reasons To Accept:**

The proposed method is simple and it shows improved generation performance and decreased inference time compared with Llama2.

**Reasons To Reject:**

1. The proposed method has only been evaluated on Llama2, leading to limited generalization.

2. Temp-Lora performs better than Llama2. This is good. But the paper only shows the performance of Llama2, lacking some strong baselines such as flashattention, pagedattention. Therefore, we can't know how good Temp-lora is compared with some strong models.

---

> ### Author Rebuttal · Authors · 2024-05-29
>
> **Concern 1**: The proposed method has only been evaluated on Llama2 & lacking some strong baselines
>
> Thanks for your valuable suggestions! Firstly we should clarify that we already compare with a **very strong baseline "Together LLaMA2-7B-32k"** in our paper. Please refer to our response to **Conern 1 from Reviewer 3Zws**, for more details of this strong baseline, and the comparision between Temp-Lora and  other existing baselines.
> 	Furthermore, please refer the response to **Concern 1  from Reviewer Sdmf** for details about the evaluation of the proposed method on other LLMs.
>
> **Concern 2**: Need some cases.
>
> We apologize, but due to the difficulty in demonstrating the effects of long-text generation, we followed the majority of work in this field and used PPL as the evaluation metric. Upon your suggestion, we conducted an additional experiment to allow you to intuitively understand Temp-Lora's effects in long-context tasks.
>
> We selected a book from PG19, BIOGRAPHIA LITERARIA, written by Samuel Taylor Coleridge, and used the content of this book as the context, taknig Llama2-7B-Chat as the base model to compare the effect of Temp-Lora.
> After we input the instruction:**'Could you please introduce four main characters that appear frequently in the book BIOGRAPHIA LITERARIA by Samuel Taylor Coleridge**, the model with Temp-Lora output:
> 'The book is divided into three parts: the first part consists of biographical sketches of various writers, including William Shakespeare, John Milton, William Wordsworth, and Alexander Pope; the second part contains critical essays on Shakespeare's plays and Milton's poetry; and the third part...'
>
> After string matching BIOGRAPHIA LITERARIA, we found that William Shakespeare, John Milton, William Wordsworth, and Alexander Pope **appeared 55, 50, 102, and 12 times respectively, indeed being the main characters in this book**.
>
> When we input the last 4K tokens of this book and the same instruction to the vanilla Llama2-7B-4K, the model's output is as follows (due to the length limit of the rebuttal, we omitted some irrelevant content):
> 'Here is a brief introduction to the main characters in Samuel Taylor Coleridge's Biographia Literaria:\n\n1. William Wordsworth ....\n2. John Wilson - ...\n3. ....\n4. Thomas De Quincey ....'
> Unfortunately, **only William Wordsworth among these four actually appears in the book**. The comparison of these two cases fully illustrates the effectiveness of Temp-Lora in long-context tasks.

---

> > ### Comment · Reviewer_1qNy · 2024-06-05
> > **Reply to authors**
> >
> > Thanks for the authors' response! I think most of my concerns have been addressed. Therefore, I increase my score to 6.
> >
> > By the way, I feel it's not a good idea to direct one reviewer to read the responses addressed to another reviewer.

---

> > > ### Author Response · Authors · 2024-06-06
> > > **Thanks**
> > >
> > > Thank you for your recognition of our work!
> > >
> > > We sincerely apologize for directing one reviewer to read the responses addressed to another reviewer. In fact, if there were no word limit for the rebuttal, we would certainly not have done so. However, due to the word limit of the COLM rebuttal (2500 characters), we were unable to fully include responses to all concerns in your rebuttal. In fact, the rebuttal provided to you has already reached 2495 characters. We resorted to this measure out of necessity, and we apologize once again for disrupting your review experience.

---

### Official Review · Reviewer_Sdmf · 2024-05-10

**Rating:** 8
**Confidence:** 3
**Ethics Flag:** 1

**Summary:**

This paper proposes that a Low Rank Adaptation module (Temp-Lora) that is temporarily invoked and discarded during the generation of long text. The results of applying this model to text generation tasks appear to be substantial, with improvements in perplexity accompanied by reductions in memory usage and inference latency.

**Questions To Authors:**

Acronyms should be spelled out when they are introduced in the abstract and introduction. The manner in which the paper introduces the terms KV cache and Lora imposes unnecessary presuppositional burdens on the reader.  The KV cache is not really an organic part of a language model, but a heuristic designed to make inference more efficient. Similarly Lora. These issues should be easily remediable before the final version is ready.

**Reasons To Accept:**

1.	Technological impact: The paper proposes replacing the widely used KV cache with a mechanism that stores context information in model parameters during training and generation and with some apparent savings of computational cost. This reviewer, at least, found the approach interesting, and potentially useful.

The Temp-Lora module only exists during long-text generation, and therefore requires a mechanism to determine when then long-text generation is being performed. At some leve, this proposal is a technically-sophisticated hack—an interesting one, but nevertheless one whose functionality might eventually be more organically folded into the model itself. In its current form, therefore, the proposal is probably not a long-term solution, but one that allows the fields to move ahead while defining a performance space that a longer-term, more-organic solution would need to cover. The definition of this achievable performance space is probably the primary reason that this paper should be accepted.

2.	Evaluation:  The paper presents extensive ablation studies showing consistently improved results at least on the metrics and datasets employed.

3.	Clarity:  The paper should be largely clear to someone who is well accomplished in language modelling.  For the most part it is readable and interesting, with good documentation of results and valuable discussion.

**Reasons To Reject:**

Evaluation:  Although the evaluation is extensive, it has limitations.  The only comparisons made in Table 1 are with models in the llama family.  The paper would be more convincing if they had shown benefits using other Open Source models, e.g., Phi-2 or Mistral, as well. (Does the efficacy of the approach depend on the model?)

it is a pity, perhaps, that the paper does not attempt to compare performance with one of more of the approaches noted in the second paragraph of the introduction. The perplexity and BLEU improvements seem large, but the reader does not get to see how the approach compares with some of the alternative approaches available.

Overreliance on a single dataset (PG19) may be an issue. The authors used a random sample of 40 books with high perplexity from within this data set. Though the reasons for this manner of selection are stated in the paper, given that the models may have seen this dataset during training, the high starting perplexity is likely to bias the results. It may be easier to obtain strong percentage-wise reductions from a higher starting point than a lower one.

---

> ### Author Rebuttal · Authors · 2024-05-29
>
> **Concern 1**: Does the efficacy of the approach depend on the model?
> 	Thank you for your suggestion! Temp-Lora significantly improves almost all other Open Source models, but we only selected the Llama family as their representative in the paper. Upon your suggestion, we present the effects of Temp-Lora on five models in the table below (Due to differences in the tokenizer, some models do not have the interval of 500K+).
>
> |Model|0-100K |100-300K |300-500K |500K+ |Avg.|
> | :--| :--|:--|:--|:--|:--|
> |mistral01-7B-8K|10.58 |10.16 |8.83 |4.32 |10.20 |
> |+TL|10.34 (2.2%) |9.57 (5.7%) |8.19 (7.1%) |3.74 (13.3%) |9.73 (4.5%) |
> |phi-1-2k|195.31 |183.33 |149.02 | |186.08 |
> |+TL|94.04 (51.8%) |67.52 (63.1%) |52.33 (64.8%) | |76.11(59.0%) |
> |phi-2-2k|22.12 |23.12 |22.76 | |22.78 |
> |+TL|22.53 (-1.8%)|21.29 (7.8%) |19.37 (14.8%) | |21.68 (4.8%) |
> |qwen-7B-8k|18.39 |17.51 |14.08 | |17.70 |
> |+TL|18.02 (2.0%) |16.44 (6.1%) |13.05 (7.3%) | |16.90 (4.5%) |
> |qwen-1.8B-8k|27.03 |25.30 |21.96 | |25.84 |
> |+TL|25.33 (6.2%) |22.23 (12.1) |18.32 (16.5%) | |23.26 (9.9%) |
>
> We will supplement these experimental results in the next version of the manuscript
>
> **Concern 2**: the paper does not attempt to compare performance with one of more of the approaches noted in the second paragraph of the introduction.
>
> Firstly we should clarify that we already compare with a **very strong baseline "Together LLaMA2-7B-32k" (refer as Llama2-7B-4k)** in our paper. Please refer to our response to Reviewer 3Zws, Conern 1, for more details of this strong baseline, and the comparision between Temp-Lora and other existing long-text generation methods.
>
> **Concern 3**: Overreliance on a single dataset (PG19) may be an issue.
>
> Yes, you are correct, this is an issue that also troubles us. One reason for this issue can be found in the **response to Concern 2 for Reviewer 3Zws**: Currently, all benchmarks, such as LongBench and Arxiv, are **too short** to validate the effectiveness of Temp-Lora. Therefore, to the best of our knowledge, we have found two datasets that are more suitable for super-long text generation scenarios: PG19 and GuoFeng. Since most of the novels in PG19 are still too short, we had to re-organize it to better validate the performance of Temp-Lora.
>
> **Concern 4**: Acronyms should be spelled out.
>
> Thank you very much for your kind suggestions! We will carefully re-organize the introduction to make this paper more easy to follow!

---

> > ### Comment · Reviewer_Sdmf · 2024-06-06
> >
> > I have read the authors' response.  I think this is a decent paper. If some of the issues can be fixed, it may go over quite well.

---

> > > ### Author Response · Authors · 2024-06-06
> > > **Thanks**
> > >
> > > Thank you for your comments!

---

### Official Review · Reviewer_3Zws · 2024-05-14

**Rating:** 5
**Confidence:** 5
**Ethics Flag:** 1

**Summary:**

In this paper, the authos proposed Temp-Lora, an alternative way for efficient long text generation. Its essence lies in training during the inference process using the generated output. It enables the storage of nearly infinite context information directly within the model’s parameters, marking a distinct difference from existing attention weights-based techniques.
The experimental results across various applications demonstrated the profound impact of Temp-Lora.

**Questions To Authors:**

1. I am interested in understanding the performance of the proposed method within the context of batch decoding. Could the authors provide a detailed analysis of its efficiency and effectiveness in such scenarios?

2. In Table 1, why does the perplexity (PPL) decrease as the range extends further back? Additionally, how is the data distributed across the different intervals? A thorough explanation of these observations would be beneficial.

**Reasons To Accept:**

1. The proposed method is interesting and has the potential to significantly enhance both effectiveness and efficiency in the domain of long-context generation.

2. This paper is well-written, with a clear and comprehensible description of the method.

**Reasons To Reject:**

1. The experimental results presented are insufficient. The authors must first conduct a comparative analysis with existing methods for long context generation. Additionally, it is imperative to validate the proposed method using established long-text benchmarks to ensure its robustness and applicability.
[1] Dynamic-NTK: https://www.reddit.com/r/LocalLLaMA/comments/14mrgpr/dynamically_scaled_rope_further_increases/
[2] LongBench: https://github.com/THUDM/LongBench

2. The primary advantage of the proposed method is its ability to operate without the need for saving the KV Cache. The authors should provide a comprehensive demonstration of the improvements in overall efficiency across various settings and scenarios to substantiate this claim, e.g., throughput in batch decoding.

---

> ### Author Rebuttal · Authors · 2024-05-29
>
> **Concern 1**: Comparision with existing methods:
>
> Thanks for your valuable suggestions! Firstly we should clarify that we already compare with a **very strong baseline "Together LLaMA2-7B-32k"** in our paper. This is the introduction of this model: https://www.together.ai/blog/llama-2-7b-32k. The
> developers followed Meta’s linear interpolation and continue pre-trained it with 1.5B tokens. We believe it is much stronger than those training-free methods. Additionally, in Table 4, we also made a comparison with the StreamingLLM method (+AS), and verified the orthogonality of Temp-Lora with it.
>
> In fact, most existing methods like Dynamic-NTK are not suitable for super-long context scenarios. This Table shows the results on PG19 after extending Llama2-7B-4K using Dynamic-NTK:
> |Model|0-100K |100-300K |300-500K |500K+ |Avg.|
> | :-:| :-:|:-:|:-:|:-:|:-:|
> | *8K | 11.2 |10.9 |9.7 |5.3 |10.9 |
> | *16K|  22.5 |24.1 |20.9 |10.5 |23.2 |
> | *24K| 65.8 |87.0 |78.3 |24.8 |77.7|
> | *32K| 106.3 |161.0 |147.1 |63.3 |137.1 |
> | *40K| 212.4 |420.6 |385.7 |145.1 |324.4 |
>
> One may find that once the context window extends to more than four times its training window, PPL will collapse directly.
>
> In addition, you may also interested in the evaluation of the proposed method on other LLMs. Please refer to **Concern 1 from Reviewer Sdmf** for details.
>
> **Concern 2**: using established benchmarks.
>
> From the introduction of LongBench: the average length of most tasks ranging from 5k to 15k.
>
> The problem is that they are **too short** to validate the effectiveness of Temp-Lora. Just as we concluded in the last sentence of section 3.3: "Note that in scenarios without extensive text, for example, less than the model’s window size in training, Temp-Lora is useless."
> 	The selected test data in this paper, PG19, is also a famous benchmark for long-text generation.
>
> **Concern 3**: a detailed analysis of batch decoding.
> 	This is the detailed memory usage and latency of Llama2-7B with batch sizes of 1, 2, and 5. The '#' symbol indicates an 'out of memory' issue.
> |Metric|Batch Size|base|24k|16K|8K|4K|2K|
> | :-:| :-:|:-:|:-:|:-:|:-:|:-:|:-:|
> |Mem. (GB)|2|64.3 |65.2 |48.3 |31.3 |22.8 |18.5 |
> |Lat.(s)|2|115.1 |117.1 |80.0 |45.9 |30.1 |29.3 |
> |Mem. (GB)|5|# |# |# |56.3 |35.2 |24.7 |
> |Lat.(s)|5| #| #| #|68.5 |49.6 |31.6 |
>
> **Concern 4**: why the PPL decrease?
>
> This is due to the natural distribution of the dataset. Due to the length limit, we will provide detailed statistics in discussion period.

---

### Author Response · Authors · 2024-06-05
**A Discussion about some common questions**

Dear reviewers! Thank you for your hard work.

During the rebuttal phase, we conducted many experiments and addressed most of your questions in detail: 1) Comparison of Temp-Lora with other existing long-text generation methods like Dynamic-NTK (Please refer to **Response to concern 1 of Reviewer 3Zws**); 2) The effect of Temp-Lora on other open-source models (Please refer to **Response to concern 1 of Reviewer Sdmf**); 3) Why we don't use benchmarks like LongBench (Please refer to **Response to concern 2 of Reviewer 3Zws**).

We are confident that after reading the rebuttal, you will have a new and more comprehensive understanding of the effectiveness of Temp-Lora. In fact, we have already applied Temp-Lora technology in many real-world applications, such as: 1) In our chat-style assistant, users can upload infinitely long documents as background information, and we will use these documents to initialize a Temp-Lora module and continue training it with subsequent user conversations. 2) We have developed some lifelong game NPCs that need to accompany users for a very long time and gradually have "unique memories" between them. These memories are implemented with Temp-Lora. 3) We also have a translation tool specifically for professional translators. They can update the Temp-Lora module with their translation history when translating long documents (such as novels) to maintain better consistency in subsequent translations (this scenario is very similar to our second experiment).

We believe that Temp-Lora represents a new paradigm for storing long-context information: in addition to being stored in the context window and external databases, model parameters can also effectively store historical knowledge. This is also consistent with human nature: people do not re-attend all context information every time they make a decision, most of the time we rely on the memory in our brains (model parameters) to make decisions, only when we need detailed information and can't remember it, will we review past records (KV cache and RAG) to help make decisions.

If you have any other questions, please send them before the end of the discussion period, and we will answer them as soon as possible.

---

### Decision · Program_Chairs · 2024-07-10

**Decision:**

Accept

**Comment:**

Some concerns were addressed by the authors during the response period.
Some reviewers are enthusiastic about the paper.
Towards Accept.